**Funding:** Yes, this project has been funded by the SHERPA University Institute via 2020-2021 annual competition to support researchers of the SHERPA University Institute and by the Social Sciences and

# Interventions facilitating access to perinatal care for migrant women without medical insurance: A scoping review protocol

**Drissa Sia[1,2,3]\*, Eric Tchouaket Nguemeleu[1,4], Idrissa Beogo[4,5], Catherine Séguin[6], Geneviève Roch[7], Janet Cleveland[8], Christina Greenaway[9]**

1 Department of Nursing, Université du Québec en Outaouais, Saint-Jérôme, Québec, Canada, 2 Département de médecine sociale et préventive de l'École de santé publique de l'Université de Montréal, Montreal, Quebec, Canada, 3 Département de gestion, d'éducation et politique de santé de l'École de santé publique de l'Université de Montréal, Montreal, Quebec, Canada, 4 École des sciences infirmières | School of Nursing. Faculty of Health Sciences. University of Ottawa, Ottawa, Ontario, Canada, 5 College of Nursing, Rady Faculty of Health Sciences, University of Manitoba, Winnipeg, Manitoba, Canada, 6 Library, Université du Québec en Outaouais, Saint-Jérôme, Quebec, Canada, 7 Faculty of Nursing, Université Laval, Quebec, Quebec, Canada, 8 McGill University Health Centre, McGill University, Montreal, Quebec, Canada, 9 Jewish General Hospital and Department of Medicine, McGill University, Montreal, Quebec, Canada

\* drissa.sia@uqo.ca

## Abstract

### Introduction

Inadequate pregnancy monitoring for pregnant migrant women without medical insurance (PMWMI) exposes them to severe complications during childbirth and consequences for the health of their child (e.g. preterm delivery, low birth weight, etc.). This scoping review aims to identify existing interventions globally to improve access to perinatal care for PMWMI. It will also highlight the strengths, weaknesses as well as the costs of these interventions.

### Materials and methods

The methodological framework developed by Arksey & O'Malley (2005) will be used. An electronic search of studies from 2000 to 2021, published in French or English, will be conducted in 12 databases. Publication in Websites of non-governmental organizations working on migrant women without medical insurance issues will be also searched. All articles related to perinatal follow-up and care of PMWMIs, regardless of design, will be included. Editorial comments will be excluded. Outcomes of interest will focus on the impacts, strengths, weaknesses, and cost of interventions. Selection of articles and data extraction will be done by two independent researchers following the Tricco et al. (2018) reporting guide. Finally, a deliberative workshop with experts will allow to identify the most promising and appropriate interventions that can facilitate access to perinatal services by PMWMIs in the Quebec province of Canada.

Humanities Research Council (SSHRC) via SSHRC institutional grants - winter 2021 competition. The funders had no role in study design, data collection and analysis, decision to publish, or preparation of the manuscript.

**Competing interests:** The authors have declared that no competing interests exist.

## Introduction

Although the phenomenon, known as the 'healthy immigrant effect', has been repeatedly reported [1, 2], in many countries [3], migrant women face inadequate pregnancy follow-up [4] resulting in more miscarriages, fetal growth problems, premature birth and high perinatal mortality [5]. Those without any medical insurance are more vulnerable than other migrants [6, 7]. They often consult a healthcare provider very late during pregnancy [8] and frequently have severe complications during childbirth [7]. They also experience a higher number of emergency caesarean sections, which can be traumatic for the mother and the unborn child [6]. These women also have a high prevalence of postpartum depression [9], parasitic [10] and other infectious diseases (Hepatitis B; Hepatitis C; HIV / AIDS). The fact they experienced challenges in accessing perinatal care [11], constitutes a double burden. This issue among pregnant migrant women without medical insurance (PMWMI) is a complex subject. Several interventions exist to address this including; (i) earlier care (ii) regular care; (iii) culturally appropriate; (iv) geographically accessible; (v) multidisciplinary; (vi) integrated with other community services / resources and at lower cost [12, 13].

We have not identified a scoping or systematic review of these interventions which would allow a better understanding of access to perinatal care for PMWMIs. It is therefore opportune to carry out some in order to list the interventions promoting the access of PMWMI women to perinatal care. As mentioned by Arksey and O'Malley (2005) and Levac (2010), a scoping review [14, 15] of these interventions will allow us to synthesize them, to know their strengths, their weaknesses and their costs. Thus, it will be possible to implement appropriate interventions depending on the context in order to facilitate access to perinatal care for PMWMIs. Knowing the costs would help inform decision-makers about the financial benefits of investing in setting up these interventions. This scoping review will fill the literature gap. It aims to identify interventions that have been employed to improve access to perinatal care for PMWMI. It will also highlight the strengths, weaknesses as well as the costs of these interventions.

## Materials and methods

### Methodological framework

The methodological framework developed by Arksey & O'Malley (2005) [14] and taken up by other authors [15, 16] will be used. This framework highlights six steps necessary for a good Scoping Review: (i) definition of the research question; (ii) identification of relevant studies; (iii) selection of studies; (iv) data extraction; (v) analysis and aggregation of results; and, (vi) consultation exercise.

### Research questions

The purpose of this scoping review project is to identify interventions (practices and policies) that support PMWMIs' access to perinatal care. It will answer the two following questions:

1. What are the interventions that facilitate access to perinatal care for PMWMI women?

2. What are their impacts on perinatal health, what are their strengths, weaknesses and what is the cost of implementation?

### Inclusion criteria

Inclusion and exclusion criteria will be based on Population, Interventions, Comparators and Design and Outcomes or Anticipated Outcomes (PICO), summarized in Table 1.

Table 1. PICO (Population, Interventions, Comparator and design, Outcomes).

| Population (P) | Women who are pregnant or in labor or in the postpartum period; 12 years and over; Migrant; without medical insurance |
|---|---|
| Interventions (I) | Carrying out pregnancy tests; Prenatal care (during pregnancy follow-up); Immediate obstetric and neonatal care (during labor, delivery and within two hours after delivery); Postpartum care (within 42 days of delivery); Newborn care (up to 28 days after birth); Pregnancy follow-up; prenatal education; vaccination; Screening for diseases during pregnancy; Childbirth assistance. |
| Comparator and design (C) | All articles, regardless of design, will be included except editorial articles and comments |
| Outcomes or intended results (O) | Strengths and weaknesses (facilitators and obstacles) of the targeted interventions; cost of these interventions and impacts / effect of these interventions |

**Population (P).** Migrants are not a homogeneous population. According to the glossary of the International Organization for Migration, the term "migrant" refers to "a person who has voluntarily moved or is forced to move, whether within a country or across an international border, temporarily or permanently, and for a variety of reasons, in order to improve his or her material and social conditions, his or her future prospects or those of his or her family" [17]. In addition to coming from different countries and cultures, a distinction is made between migrants: (i) in a regular situation whose entry and stay in the host country are in accordance with the applicable law; (ii) and, in an irregular situation contravening the regulations of the host country by entering it irregularly or remaining there beyond the validity of the residence permit [17, 18]. As part of this review, internal displaced as well as refugees and asylum seekers without health insurance will also be considered. This review will include articles that focus on migrant women 12 years and older without health insurance who are either pregnant, in labor, or postpartum. Articles that focus on the perinatal period of migrant women without health insurance will be considered.

**Interventions (I).** They include the care provided to uninsured migrants during the perinatal period as well as the policies and practices implemented to promote their access to this care. The perinatal period adopted for this study is broader than the World Health Organization definition [19] and extends from the beginning of pregnancy to the first months of life of the newborn [20]. Care during pregnancy including prenatal education, labor, delivery and postpartum, as presented in the Médecin du Monde (MdM) reference framework [21] will therefore be considered. Details of prenatal care (during pregnancy follow-up), immediate obstetric and neonatal care (during labor, delivery, and within two hours of delivery), postpartum care (within 42 days of delivery), and newborn care (up to 28 days after birth) are presented in Table 2.

**Comparator and design (C).** All empirical scientific studies, regardless of design, in French or English of any high, medium or low income country- (see S1 Appendix which presents the complete list of countries as listed by the World Bank in 2021) [22] will be accepted. Media articles, editorial comments, as well as studies focusing only on the profile of PMWMIs will be excluded. No comparison is foreseen.

**Outcomes or intended results (O).** The impacts, strengths, weaknesses and implementation costs of interventions aimed at providing access to perinatal care for PMWMIs will be reported.

## Data sources and study identification

This scoping review has been registered in Research Registry (6864; https://www.researchregistry.com/browse-the-registry#home/). The recommendations of the Preferred

**Table 2.  Types of intervention / care and related essential intervention package.**

| Type of intervention | Package of essential interventions |
|---|---|
| **Prenatal care** (provided during pregnancy by a qualified health professional to ensure the birth of a healthy child with minimal risk to the mother) | History and clinical examination |
| | Management of unwanted pregnancies |
| | Management and/or referral of pregnant women with complications |
| | Information and counseling (health education) |
| | Preparation for childbirth |
| | The childbirth preparation plan |
| | Prevention and management of anemia |
| | Tetanus vaccination |
| | Prevention and treatment of malnutrition |
| | Screening and treatment of syphilis |
| | Information, screening and treatment of precancerous cervical cancer lesions |
| | Identification of victims of gender-based violence |
| | Screening and management of pregnancy complications |
| | Registration of medical data |
| | Malaria prevention and management [a] |
| | Antiparasitic treatment [a] |
| | Prevention of mother-to-child transmission of HIV [a] |
| **Immediate obstetric and neonatal care** (preventive and curative care provided during labor, delivery, and the immediate postpartum period (2 hours after delivery) to reduce maternal and neonatal mortality and morbidity through early detection and timely management of obstetric and neonatal complications) | Early referral for complications and/or situations requiring specialized care |
| | Care during labor and delivery in the presence of a qualified health professional |
| | Neonatal care |
| | Immediate postpartum (maternal and newborn monitoring, information and counseling on hygiene, home health care, nutrition, exclusive breastfeeding, family planning, postpartum care, child care, and danger signs and emergency preparedness) |
| | Detection and management of victims of gender-based violence |
| | Prevention and early detection of fistula |
| | Registration of births and/or deaths at the civil registry and of medical information in health registers and diaries |
| | Monitoring and response to maternal deaths and near misses "échappées belles, here Fin French at the institutional and community levels |
| | Prevention of malaria [a] |
| | Prevention of mother-to-child transmission of HIV [a] |
| **Postpartum care** (given to the mother after delivery and up to 42 days after delivery (6 weeks), the postpartum period is divided into three parts: (i). the immediate postpartum during the first 24 hours of the newborn's life; (ii) the early postpartum from the 2nd to the 7th day after birth; (iii) the late postpartum covering a period from the 8th to the 42nd day after birth). | Monitoring of the mother's health and well-being |
| | Screening and management of postpartum complications |
| | Information and counseling |
| | Promotion, protection and support of exclusive breastfeeding |
| | Prevention and management of anemia |
| | Tetanus vaccination |
| | Screening and treatment of STIs |
| | Attention to victims of gender-based violence |
| | Family planning counseling and provision of appropriate contraceptive methods |
| | Early detection of fistula |
| | Registration of births, deaths, and medical information in registers and health books |
| | Monitoring and responding to maternal deaths and "échappées belles" |
| | Prevention of mother-to-child transmission of HIV |

(*Continued*)

**Table 2.** (Continued)

| Type of intervention | Package of essential interventions |
|---|---|
| **Neonatal care** (Provided to children after birth within the first 28 days to ensure a smooth transition to extra-uterine life) | Resuscitation of the newborn at birth |
| | Immediate care of the newborn |
| | Promotion, early initiation |
| | Monitoring and surveillance of the newborn's condition and well-being |
| | Hepatitis B, BCG and polio vaccinations |
| | Screening and initial treatment of at-risk newborns |
| | Prevention and management of congenital syphilis |
| | Information and counseling |
| | Registration of births and deaths at the civil registry, and medical information in health registers and health cards |
| | Prevention of malaria [a] |
| | Prevention of mother-to-child transmission of HIV [a] |

[a]: To be considered in endemic areas

Reporting Items for Systematic Reviews and Meta-analyses Extension for Scoping Reviews (PRISMA-ScR) [23] (see S1 Checklist) will be followed in carrying out this review. The specifications of the elements relating to the construction of the Flow diagram will be explicitly mentioned. Articles will be selected via: (i) electronic bibliographic databases CINHAL, Web of Science, Medline-Ovid, Pubmed, Embase, Cochrane Library, Scopus, ScienceDirect, Hinari, Lilacs, Cairn and Banque de Données Santé Publique (BDSP); (ii) reference lists; (iii) key journals in the field of immigration (Revue Migrations Forcées; Migrations Société); and (iv) Websites of non-governmental organizations (Médecin du monde, Médecin sans Frontières (MSF), United Nations High Commissioner for Refugees (UNHCR). A working meeting between the co-researchers of this project, including an experienced librarian (CS), has already allowed the definition of the search strategy presented in Table 3. This strategy was developed using descriptors or thesauri with the logical operators "AND" and "OR" to identify relevant studies published between 2000 and 2021 in French or English. The year 2000 was a landmark turning point for the entire world with the formulation of the eight Millennium Development Goals (MDGs), where emphasis was put in poverty and hunger reduction (MDG 1) the promotion of gender equality and empower of women (MDG 3) and among other child mortality reduction (MDG 4) [24]. The selected articles will be exported to Rayyan [25] via EndNote.

## Study selection

A librarian of the Université du Québec en Outaouais (CS) will be responsible for applying the search strategy and extracting articles from the databases in order to prepare the EndNote bibliographic database. This database will then be cleaned and duplicates removed before exporting the articles to Rayyan platform. The selection of articles will be based on the inclusion and exclusion criteria defined by the PICO and will be done in two stages. At the first stage of selection, an article will be eligible if, through its title and abstract, it is possible to clearly identify the PICO in relation to those selected for the study. To do this, two researchers will independently examine the title and the abstract of the articles identified according to an algorithm built with predetermined eligibility criteria (Fig 1) and will justify in writing the eligibility of the articles. An article will be retained if both declare it eligible. Any disagreement will be

**Table 3. Search strategy in CINAHL, to be modified as needed for other databases.**

| Number | Queries |
|---|---|
| 1 | TI (Immigrant* OR migrant* OR temporary worker* OR 'migrant worker*' OR 'foreign worker' OR 'foreign workers' OR 'domestic worker*' OR 'live-in caregiver*' OR caregiver OR caregivers OR refugee OR refugees* OR asylum) OR AB (Immigrant* OR migrant* OR temporary worker* OR 'migrant worker*' 'foreign worker' OR 'foreign workers' OR 'domestic worker*' OR 'live-in caregiver*' OR caregiver OR caregivers OR refugee OR refugees* OR asylum) |
| 2 | (MM "Immigrants+") OR (MM "Refugees+") |
| 3 | 1 OR 2 |
| 4 | TI (Woman OR women OR female OR females OR adolescent OR adolescents) OR AB (Woman OR women OR female OR females OR adolescent OR adolescents) |
| 5 | 3 AND 4 |
| 6 | TI (parturition OR pregnancy OR pregnancies OR pregnant OR gestat* OR Prenatal OR 'Pre natal' OR 'Pre-natal OR ante-natal OR 'ante natal' OR antenatal OR 'peri natal' OR perinatal OR peri-natal OR 'post natal' OR post-natal OR postnatal OR 'newborn care' OR 'neonatal care') OR AB (parturition OR pregnancy OR pregnancies OR pregnant OR gestat* OR Prenatal OR 'Pre natal' OR 'Pre-natal OR ante-natal OR 'ante natal' OR antenatal OR 'peri natal' OR perinatal OR peri-natal OR 'post natal' OR post-natal OR postnatal OR 'newborn care' OR 'neonatal care') |
| 7 | TI (illness OR illnesses OR disease OR diseases OR ailed OR suffering OR pain) OR AB (illness OR illnesses OR disease OR diseases OR ailed OR suffering OR pain) |
| 8 | 6 OR 7 |
| 9 | TI ('Childbirth assistance' OR 'delivery assistance' OR 'assisted delivery' OR 'Assisted Parturition' OR 'Assisted childbirth) OR AB ('Childbirth assistance' OR 'delivery assistance' OR 'assisted delivery' OR 'Assisted Parturition' OR 'Assisted childbirth) |
| 10 | (MH "Prenatal Care") |
| 11 | TI (monitoring OR 'follow up' OR exam* OR visit OR 'education OR treatment OR 'follow up' OR follow-up OR Screening OR screen OR test OR tests OR testing OR diagnosis OR care OR consultation OR service OR visit OR diagnosis) OR AB (monitoring OR 'follow up' OR exam* OR visit OR 'education OR treatment OR 'follow up' OR follow-up OR Screening OR screen OR test OR tests OR testing OR diagnosis OR care OR consultation OR service OR visit OR diagnosis) |
| 12 | TI (Ceasarean OR caesarian OR section OR cesarian OR "cesarean section" OR "caesarean section" OR CS OR C-S OR "C Section" OR C-Section OR "abdominal delivery" OR "abdominal deliveries" OR "cesarean birth" OR "cesarian birth" OR "cesarean deliveries" OR "cesarian deliveries" OR "cesarean delivery" OR "cesarian delivery") OR AB (Ceasarean OR caesarian OR section OR cesarian OR "cesarean section" OR "caesarean section" OR CS OR C-S OR "C Section" OR C-Section OR "abdominal delivery" OR "abdominal deliveries" OR "cesarean birth" OR "cesarian birth" OR "cesarean deliveries" OR "cesarian deliveries" OR "cesarean delivery" OR "cesarian delivery") |
| 13 | (MH "Childbirth") OR (MH "Childbirth Education") |
| 14 | (MM "Delivery, Obstetric+") |
| 15 | 9 OR 10 OR 11 OR 12 OR 13 OR 14 |
| 16 | TI (Uninsured OR insurance OR 'without health insurance' OR 'Uninsured patient*') OR AB (Uninsured OR insurance OR 'without health insurance' OR 'Uninsured patient*') |
| 17 | (MM "Insurance, Health+") |
| 18 | 16 OR 17 |
| 19 | 5 AND 8 AND 15 AND 18 |

resolved either by discussion, or by another co-researcher who will review the title and abstract and give a verdict. The article will be retained if two of these three people declare it eligible. This is an iterative process in which the researchers involved in the project will meet at the beginning, middle and end of the abstract review phase to discuss challenges and uncertainties related to the selection of studies in order to refine the research strategy if necessary. At the final stage, eligible articles will be read in their entirety. For any question regarding the content of an article, the authors will be contacted for further clarification. To harmonize the selection process, the selection of 10% of the articles will be discussed by all researchers beforehand.

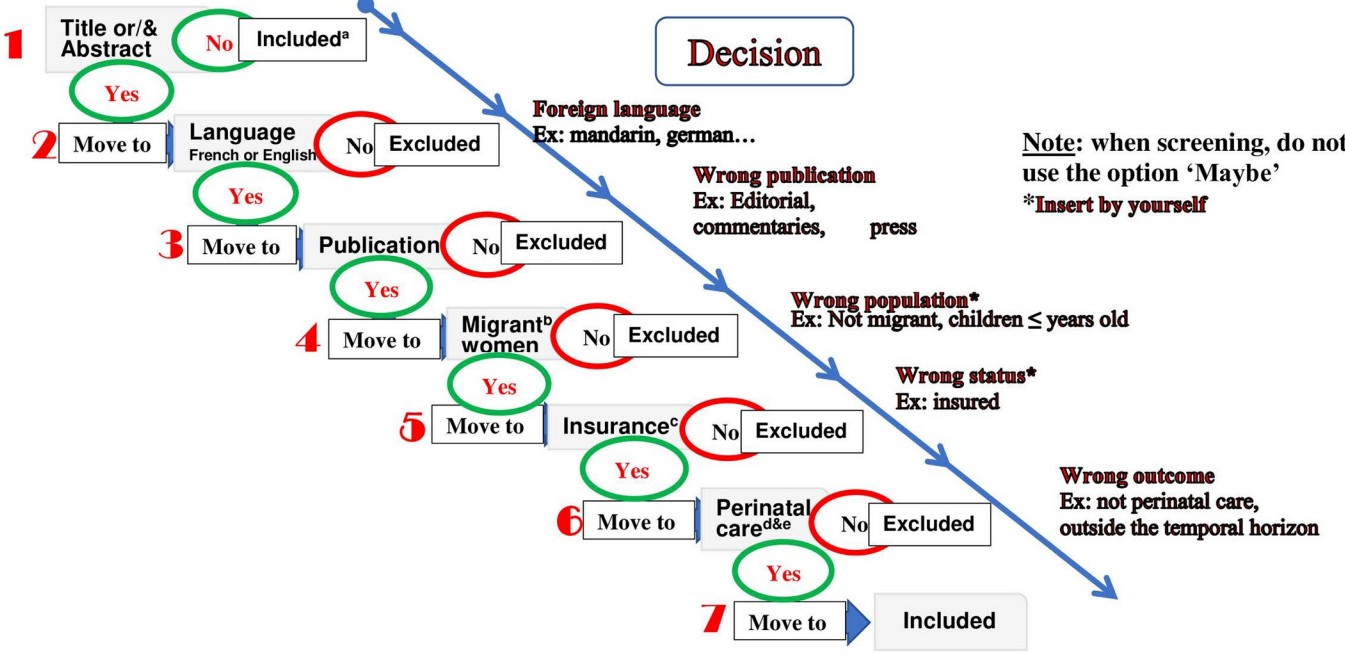

**Figure 1: First round screening algorithm of a peer- reviewed paper**

**Legend:**
[a] *The plain text twill be considered for reading;*
[b] Migrant women: immigrant, refugees, temporary worker, foreign worker, caregiver, asylum seeker
*EXCLUDED*: men, children under 12 year old;
[c] **Insurance: any type of health insurance**
*EXCLUDED*: those having any type health insurance;
[d] **Perinatal care:** cares offered (ex.: visit, treatment, test, vaccination) during the pregnancy or the labor or in postnatal
*EXCLUDED*: Studies assessing technology **or** studies purely clinically oriented;
[e] **Perinatal care [Time horizon]:** include all services and care sought for healthcare system from day1 of the pregnancy
to the first anniversary of the child (11 month postbirth)
*EXCLUDED*: outside this time period;
The numbers 1,2,3…7 are compulsory consecutive steps of the screening flow;
Colors used have no special meaning.

**Fig 1. First round screening algorithm of a peer-reviewed paper.**

## Data extraction

We developed a data extraction grid (S2 Appendix) that is an adaptation of the grid used by Stirling Cameron et al. [26] in their scoping review on access to and use of sexual and reproductive health services by refugee and asylum-seeking women in high-income countries. This grid, built in an excel spreadsheet, will allow the extraction of the following information: name of authors; year of publication; country; title and abstract; study design; study population; sample size; interventions (antenatal care, immediate obstetric and neonatal care, postpartum care, newborn care, policies and practices in favor of this care); impact; strengths; weaknesses; and cost of implementation of these interventions. Because the data extraction process is iterative, the data extraction grid will be updated by the researchers as the data extraction progresses. Two researchers will independently extract data from the first five studies, and then meet to harmonize the data extraction (consistent with the research question and purpose of the study).

## Data analysis

The data analysis will be done in four steps: 1) organizing the studies into logical categories related to the targeted objectives; 2) reporting the results by intended objectives; 3) examining

their meaning; 4) comparing the interventions implemented and their outcomes in regard to the setting (high-, medium- or low-income country), and 5) discussing the implications for future research, clinical practice, and health policy. In addition, a narrative summary will describe how the results are related to the research questions. The results that will be obtained, i.e., the interventions in favor of access to perinatal care by PMWMIs, the impact, strengths, weaknesses, and cost of implementation of these interventions, will be submitted for expert review as described in the following section.

## Expert consultation

This process is based on a participatory approach [27] in order to deepen the results [15] and to constructively contextualize [28] the identified interventions with expert researchers and practitioners whose work focuses on access to sexual and reproductive health services for vulnerable persons and immigrant women. In addition to the researchers, 15 experts who will take part in this workshop will have the following profiles: (i) academics (three) working in the field of the access to perinatal care for vulnerable populations; (ii) experts (three) of a non-governmental organization (NGO) that provide perinatal care to vulnerable populations; (iii) experts (three) of community-based organizations that provide perinatal care to PMWMI; (iv) experts (three) who work or have worked in a municipality's immigration team; (v) experts (three) of the public health system and who are familiar with the issues of access to perinatal services for PMWMI. Experts will be identified and invited by email to participate in a workshop. Upon their agreement, written informed consent will be obtained prior to the start of the workshop. Due to the context of the COVID-19 pandemic, the deliberative workshop will be virtual via a Zoom platform. It will address the following questions: (i) What interventions would promote access to perinatal services for PMWMI in the Quebec context? (ii) Which are the most promising and affordable (efficient) for this population in Quebec? (iii) What recommendations (for policy makers, community organizations, NGOs) could be made based on the interventions identified? The deliberative workshop will take approximately three hours. It will be conducted in four stages: (1) The first (30 minutes) will be the presentation of the results of the scoping review by the research team; (2) The second (45 minutes) will be group work, four groups of 5–6 people. Each group will discuss in depth the questions posed and summarize them; (3) The third (60 minutes) will consist of a plenary presentation of the conclusions of the group work; (4) Finally, the research team will synthesize the recommendations and present them in plenary for final validation (30 minutes).

This project has been accepted by the Research Ethics Committee of the Université du Québec en Outaouais. Written informed consent will be required from the experts who will be joining the consultation session.

## Discussion

Our project aims to identify interventions that support access to perinatal care for pregnant migrant women without medical insurance by answering broad research questions and providing an overview of the literature. Thus, a scoping review described above is more appropriate for this study than the more common systematic review. Also, by using this approach we can determine whether it is appropriate to undertake a full systematic review. Another strength of this approach is the fact that it allows, through the consultation of experts, to deepen and adapt the results obtained [29].

Although the results of this scoping review should be interpreted with caution as only French or English studies will be considered and their quality is not assessed. In addition, a scoping review focuses on the mapping and scope of studies, rather than the depth of

information; it describes what is known rather than providing new knowledge. These limitations are inherent in this method and do not affect the results that will be obtained. At the end of the expert consultation, the most promising and appropriate interventions will be identified to facilitate access to perinatal services by PMWMIs in the Quebec province of Canada. In addition to a scientific article, an advocacy and policy brief will also be drafted and submitted to policy makers in charge of immigration and migrant health to facilitate the consideration and integration of the results obtained.

## Supporting information

**S1 Checklist. Preferred reporting items for systematic reviews and meta-analyses extension for scoping reviews (PRISMA-ScR) checklist.**
(DOCX)

**S1 Appendix. Complete list of countries considered.**
(DOCX)

**S2 Appendix. Extraction grid.**
(DOCX)

## Author Contributions

**Conceptualization:** Drissa Sia, Eric Tchouaket Nguemeleu, Idrissa Beogo.

**Funding acquisition:** Drissa Sia.

**Investigation:** Drissa Sia, Eric Tchouaket Nguemeleu, Idrissa Beogo, Geneviève Roch.

**Methodology:** Drissa Sia, Eric Tchouaket Nguemeleu, Idrissa Beogo, Catherine Séguin, Geneviève Roch, Janet Cleveland, Christina Greenaway.

**Project administration:** Drissa Sia.

**Validation:** Drissa Sia, Eric Tchouaket Nguemeleu, Idrissa Beogo, Catherine Séguin, Geneviève Roch, Janet Cleveland, Christina Greenaway.

**Writing – original draft:** Drissa Sia.

**Writing – review & editing:** Drissa Sia, Eric Tchouaket Nguemeleu, Idrissa Beogo, Catherine Séguin, Geneviève Roch, Janet Cleveland, Christina Greenaway.

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
