## [Decision Letter · Decision Letter 0]

22 Oct 2021

PONE-D-21-27767Interventions facilitating access to perinatal care for migrant women without medical insurance: a scoping review protocolPLOS ONE

Dear Dr. Sia,

Thank you for submitting your manuscript to PLOS ONE. After careful consideration, we feel that it has merit but does not fully meet PLOS ONE’s publication criteria as it currently stands. Therefore, we invite you to submit a revised version of the manuscript that addresses the points raised during the review process. Please review and address all comments raised by the reviewers.

We look forward to receiving your revised manuscript.

Kind regards,

Kelli K Ryckman

Academic Editor

PLOS ONE

Journal Requirements:

“Yes, this project has been funding by the SHERPA University Institute via 2020-2021 annual competition to support researchers of the SHERPA University Institute and by the Social Sciences and Humanities Research Council (SSHRC) via SSHRC institutional grants - winter 2021 competition”          

“Acknowledgements:

We would like to thank the SHERPA University Institute and The Social Sciences and Humanities Research Council (SSHRC) for funding the project.

Funding:

This project has been funding by the SHERPA University Institute via 2020-2021 annual competition to support researchers of the SHERPA University Institute and by the Social Sciences and Humanities Research Council (SSHRC) via SSHRC institutional grants - winter 2021 competition.”

We note that you have provided additional information within the Acknowledgements Section. Please note that funding information should not appear in the Acknowledgments section or other areas of your manuscript. We will only publish funding information present in the Funding Statement section of the online submission form.

 “Yes, this project has been funding by the SHERPA University Institute via 2020-2021 annual competition to support researchers of the SHERPA University Institute and by the Social Sciences and Humanities Research Council (SSHRC) via SSHRC institutional grants - winter 2021 competition”

5. We note that you have referenced (Open Grey, Wholis, unpublished reports]) which has currently not yet been accepted for publication. Please remove this from your References and amend this to state in the body of your manuscript: (“Open Grey, Wholis, [unpublished reports]”) as detailed online in our guide for authors

8. We note that this manuscript is a systematic review or meta-analysis; our author guidelines therefore require that you use PRISMA guidance to help improve reporting quality of this type of study. Please upload copies of the completed PRISMA checklist as Supporting Information with a file name “PRISMA checklist”.

Reviewers' comments:

Reviewer's Responses to Questions

**Comments to the Author**

1. Does the manuscript provide a valid rationale for the proposed study, with clearly identified and justified research questions?

Reviewer #1: Yes

Reviewer #2: Yes

2. Is the protocol technically sound and planned in a manner that will lead to a meaningful outcome and allow testing the stated hypotheses?

Reviewer #1: Yes

Reviewer #2: Partly

3. Is the methodology feasible and described in sufficient detail to allow the work to be replicable?

Reviewer #1: Yes

Reviewer #2: No

4. Have the authors described where all data underlying the findings will be made available when the study is complete?

Reviewer #1: Yes

Reviewer #2: Yes

5. Is the manuscript presented in an intelligible fashion and written in standard English?

Reviewer #1: Yes

Reviewer #2: Yes

6. Review Comments to the Author

You may also provide optional suggestions and comments to authors that they might find helpful in planning their study.

Reviewer #1: This is a well written scoping review proposal to identify interventions to improve access to perinatal care for migrant women without medical insurance.

A clear structure for the review is given, with a comprehensive search strategy, inclusion and exclusion criteria given.

I have one main critique:

Within the data analysis section it states in step 4 that they will be: “discussing the implications for future research, clinical practice, and health policy.” The methodological framework for scoping reviews developed by Arksey & O’Malley (2005) specifically states that due to the quality of studies not being assessed within a scoping review that as a result the generalisability and robustness of the studies cannot be determined. Careful consideration therefore needs to be given to the extent to which their scoping review results will lead to implications for policy and practice, or further justification given if they do feel this is appropriate.

All other comments are of a very minor nature:

The sentence in the introduction: “These women also have a high prevalence of postpartum depression6, parasitic7 and other infectious diseases (Hepatitis B; Hepatitis C; HIV) / AIDS) due to the challenges experienced in accessing perinatal care8.” Currently reads as though women are at risk of infectious diseases due to the challenges of accessing perinatal care, but my understanding is that these infections are already present but make accessing care early all the more necessary.

No explanation/justification is given as to why the search begins from the date 2000.

If there are any questions regarding the content of an article, will the article authors be contacted in any way for clarification?

Please ensure correct ordering of Tables / Appendices – ie in the text Table 3 is referred to prior to Table 2 and Appendix 2 prior to Appendix 1 – please reorder correctly.

Please note the abbreviation is incorrect in several places – ie PMWMI has been replaced by the incorrect PMWNI (seen in abstract and introduction – but please check throughout).

Table 3: In English “échappées belles" are normally termed “near misses”. I had to Google for a translation to the above term, so it could help some readers if both terms were used within the table.

Appendix 1: Is it possible to rotate the table by 90 degrees for publication, so that the columns become rows to make it easier for the reader to see what data will be extracted.

Appendix 2: Please add the World Bank reference from the text to the Appendix as well

Figure 1: In the legend it states “4: Time horizon ...” – however this is not added into the actual diagram at present.

Reviewer #2: The authors of this scoping review are seeking to addresses an important issue, and their overall approach is well-suited to the topic of inquiry. It will make an important contribution to the literature on maternity care for migrant women, and it is strengthened by the element of expert consultation which is included. However, the protocol is unclear in places and would benefit from further refining. I would suggest the following points are considered:

Major issues

1. p.3 – In the opening paragraph, the authors make reference to challenges faced by migrant women accessing maternity care in high-income countries. However, listed in Appendix 2 it is stated that both high-income countries as well as low- and middle-income countries will be included in this review, which is a little confusing for the reader and needs to be clarified. In addition, it will be helpful for the authors somewhere in the paper to discuss how they will deal with the differences in findings from the varying contexts of HICS’s vs LMIC’s.

2. p.3 – The introduction paints a picture of catch-all poor perinatal outcomes for migrant women. As the authors correctly acknowledge, migrant women constitute a heterogeneous group and the research is equivocal about obstetric outcomes for some migrant women, citing the healthy migrant effect. A more nuanced introduction would strengthen the introductory section.

3. p.4 – In the ‘population’ paragraph, the authors quote the definition of ‘migrant’ which describes voluntary migration, but then later on state that studies which include asylum seekers will be considered. Although migrant categories are hard to define and overlapping, for the purpose of this scoping review, it would be helpful to clearly outline what the authors are referring to by ‘migrant women’ – is this all categories of migrants, whether forced or voluntary, as the search terms suggest? If so, the definition of ‘migrant women’ which the authors cite needs adapting. Does it include those who migrate within their country (as the International Organization for Migration definition includes)?

4. p.4 – In the ‘intervention’ paragraph, the authors discuss the intervention in the PICO framework as being the actual maternity or newborn care which women receive. However, on p.3 the research question states that the authors are interested in “interventions that facilitate access to perinatal care”. There needs to be further clarity as to whether it’s perinatal care itself or interventions to facilitate access to perinatal care that is the topic of inquiry, which then needs to be reflected in the protocol title and the intervention.

5. p.4 – Newborn care is listed in the ‘intervention’ paragraph but related search terms are not included in the search strategy.

6. p.4 – In the ‘comparator and design’ section, the authors state the study inclusion criteria. From my understanding of this scoping review and according to the PICO framework, there is no ‘comparator’ in this review, so this needs to be reworked.

7. p.4/5 – In the ‘data sources’ paragraph, it is stated that grey literature will be included, but earlier on p.4 in the ‘comparator and design’ section, the authors state that all empirical scientific studies will be included – it would help the reader to understand the inclusion criteria if the authors state whether they are only going to include peer-reviewed published studies or studies that are not peer-reviewed.

Minor issues

p.3 – In the second paragraph of the introduction, the authors state that using a scoping review will allow them to synthesise the interventions, and yet there is not further mention of data synthesis in the paper. Arksey and O’Malley (2005) do not require data synthesis in their approach to scoping reviews – this just needs to be clarified in the manuscript.

This an important piece of work to conduct, and with the above revisions / clarifications, I believe it will lead to a scoping review which will contribute to the corpus of knowledge relating to maternity care for migrant women. However, based on the revisions / clarifications needed, I suggest that this protocol needs major revisions.

7. PLOS authors have the option to publish the peer review history of their article (what does this mean?). If published, this will include your full peer review and any attached files.

Reviewer #1: No

Reviewer #2: **Yes: **Esther Sharma

---

## [Author Response · Author response to Decision Letter 0]

15 Dec 2021

We already uploaded response to reviewers file.

---

## [Decision Letter · Decision Letter 1]

12 Jan 2022

PONE-D-21-27767R1Interventions facilitating access to perinatal care for migrant women without medical insurance: a scoping review protocolPLOS ONE

Dear Dr. Sia,

Thank you for submitting your manuscript to PLOS ONE. After careful consideration, we feel that it has merit but does not fully meet PLOS ONE’s publication criteria as it currently stands. Therefore, we invite you to submit a revised version of the manuscript that addresses the points raised during the review process. Please address the reviewer's additional comments in your revised submission. 

We look forward to receiving your revised manuscript.

Kind regards,

Kelli K Ryckman

Academic Editor

PLOS ONE

Journal Requirements:

Reviewers' comments:

Reviewer's Responses to Questions

**Comments to the Author**

1. Does the manuscript provide a valid rationale for the proposed study, with clearly identified and justified research questions?

Reviewer #1: Yes

2. Is the protocol technically sound and planned in a manner that will lead to a meaningful outcome and allow testing the stated hypotheses?

Reviewer #1: Yes

3. Is the methodology feasible and described in sufficient detail to allow the work to be replicable?

Reviewer #1: Yes

4. Have the authors described where all data underlying the findings will be made available when the study is complete?

Reviewer #1: Yes

5. Is the manuscript presented in an intelligible fashion and written in standard English?

Reviewer #1: Yes

6. Review Comments to the Author

You may also provide optional suggestions and comments to authors that they might find helpful in planning their study.

Reviewer #1: Thank you for the opportunity to re-review this article. There are still a couple of areas that require clarification before publication to ensure consistency throughout the manuscript.

1. There still seems to be some confusion over the inclusion of grey literature. The abstract states “Grey literature will also be searched” and the search strategy states that “(iv) sites of non-governmental organizations (Médecin du monde, Médecin sans Frontières (MSF), United Nations High Commissioner for Refugees (UNHCR)” will be searched, which would be expected to find grey literature. However, the response to the reviewers states that grey literature has been removed from the data sources. Please ensure the whole manuscript is consistent regarding this, making it explicit whether unpublished studies will be included.

2. The study selection currently appears to read that if the PICO for your review is not clearly identifiable in the title / abstract that you will exclude the study. Within a review it is usual to retain ‘potentially eligible’ studies at this stage too, where aspects of your inclusion/ exclusion criteria for the review are not fully reported in the title / abstract of the retrieved citation but may be reported if the full article is read. The flow chart suggests that studies will be included if the abstract does not provide adequate detail – but the prose needs to reflect this too. eg as stated by Askey and O’Malley (2005) “If the relevance of a study is unclear from the abstract, then the full article will be obtained.”

3. Expert consultation section – states that informed consent will be obtained, but in the ethics declaration it states that consent is not required. Could the ethics statement be updated to better reflect that while consent is not required for the review, consent will be gained for the expert consultation workshops? This will provide better consistency through the manuscript.

4. In the version provided Figure 1 does not states “ye” not “yes” and “exclu” not “exclude”. I presume this would be rectified if we had a copy of the actual Figure?

5. The definition of migrant no longer reads well – it would be better as "a person who has

voluntarily moved or is forced to move, whether within a country or across an international border, temporarily or permanently, and for a variety of reasons …”

6. I agree with the other reviewer that there is lack of clarity over the word ‘synthesize’. The authors need to give a plan for how data will be ‘synthesized’ within the expert consultation. Alternatively, would the word ‘summarize’ better describe the planned process that the word ‘synthesize’? This would also potentially more closely reflect the approach proposed by Askey and O’Malley (2005).

7. PLOS authors have the option to publish the peer review history of their article (what does this mean?). If published, this will include your full peer review and any attached files.

Reviewer #1: No

---

## [Author Response · Author response to Decision Letter 1]

14 Feb 2022

Kelli K Ryckman

Academic Editor

PLOS ONE

Dear Kelli K Ryckman

We would like to thank the reviewers for their thorough critics and comments concerning our manuscript " Interventions facilitating access to perinatal care for migrant women without medical insurance: a scoping review protocol " (PONE-D-21-27767). The manuscript has been revised according to the journal requirements and to comments of our reviewers and we have addressed all their questions.

Our answers are written in italics under each comment made by the reviewers and the changes related to these comments are presented in the text of the article, with track changes.

I. Journal Requirements:

Response: Reference list has been revised and no change made.

II. Reviewer #1: 

Thank you for the opportunity to re-review this article. There are still a couple of areas that require clarification before publication to ensure consistency throughout the manuscript.

1. There still seems to be some confusion over the inclusion of grey literature. The abstract states “Grey literature will also be searched” and the search strategy states that “(iv) sites of non-governmental organizations (Médecin du monde, Médecin sans Frontières (MSF), United Nations High Commissioner for Refugees (UNHCR)” will be searched, which would be expected to find grey literature. However, the response to the reviewers states that grey literature has been removed from the data sources. Please ensure the whole manuscript is consistent regarding this, making it explicit whether unpublished studies will be included.

Response: Thanks for the comment. We will be considering only publications from the three main non-governmental organizations involved in addressing issues concerning migrant women without medical insurance, namely Médecin du monde, Médecin sans Frontières (MSF), United Nations High Commissioner for Refugees (UNHCR). There for we will not search into any other sources except the aforementioned.

Through out the manuscript, our consideration in terms of data sources include the 12 databases and three websites.

2. The study selection currently appears to read that if the PICO for your review is not clearly identifiable in the title / abstract that you will exclude the study. Within a review it is usual to retain ‘potentially eligible’ studies at this stage too, where aspects of your inclusion/ exclusion criteria for the review are not fully reported in the title / abstract of the retrieved citation but may be reported if the full article is read. The flow chart suggests that studies will be included if the abstract does not provide adequate detail – but the prose needs to reflect this too. eg as stated by Askey and O’Malley (2005) “If the relevance of a study is unclear from the abstract, then the full article will be obtained.”

Response: Sorry for the misunderstanding. In order to light more the reader, we added in the legend of figure 1 an explanation as follow: “The plain text twill be considered for reading”. This means that any inclusion at that stage remains potentially eligible and then will be reassessed in reading the plain text.

3. Expert consultation section – states that informed consent will be obtained, but in the ethics declaration it states that consent is not required. Could the ethics statement be updated to better reflect that while consent is not required for the review, consent will be gained for the expert consultation workshops? This will provide better consistency through the manuscript.

Response: We thank you for this important comment. We formulate the following excerpt: Written informed consent will be required from the experts who will be joining the consultation session. Please, see on P 7.

4. In the version provided Figure 1 does not states “ye” not “yes” and “exclu” not “exclude”. I presume this would be rectified if we had a copy of the actual Figure?

Response: Thanks. The wording has been adjusted in the algorithm in Figure 1. 

5. The definition of migrant no longer reads well – it would be better as "a person who has

voluntarily moved or is forced to move, whether within a country or across an international border, temporarily or permanently, and for a variety of reasons …”

Response: Thanks a lot for the suggestion. We truly found it very readable. So, we take it into account. Please see on P 4. 

6. I agree with the other reviewer that there is lack of clarity over the word ‘synthesize’. The authors need to give a plan for how data will be ‘synthesized’ within the expert consultation. Alternatively, would the word ‘summarize’ better describe the planned process that the word ‘synthesize’? This would also potentially more closely reflect the approach proposed by Askey and O’Malley (2005).

Response: Thank you. As suggested, we change the word ‘synthezise’ to ‘summarize’. Please see on P 7

---

## [Editor Report · Decision Letter 2]

28 Feb 2022

Interventions facilitating access to perinatal care for migrant women without medical insurance: a scoping review protocol

PONE-D-21-27767R2

Dear Dr. Sia,

We’re pleased to inform you that your manuscript has been judged scientifically suitable for publication and will be formally accepted for publication once it meets all outstanding technical requirements.

Kind regards,

Kelli K Ryckman

Academic Editor

PLOS ONE
---

## [Editor Report · Acceptance letter]

4 Mar 2022

PONE-D-21-27767R2 

Interventions facilitating access to perinatal care for migrant women without medical insurance: a scoping review protocol 

Dear Dr. Sia:

I'm pleased to inform you that your manuscript has been deemed suitable for publication in PLOS ONE. Congratulations! Your manuscript is now with our production department. 

Kind regards, 

on behalf of

Dr. Kelli K Ryckman 

Academic Editor

PLOS ONE